

# Deconfined quantum criticality in the long-range, anisotropic Heisenberg chain

Anton Romen⋆, Stefan Birnkammer and Michael Knap

Technical University of Munich, TUM School of Natural Sciences,
Physics Department, 85748 Garching, Germany
Munich Center for Quantum Science and Technology (MCQST),
Schellingstr. 4, 80799 München, Germany

⋆ anton.romen@tum.de

## Abstract

Deconfined quantum criticality describes continuous phase transitions that are not captured by the Landau-Ginzburg paradigm. Here, we investigate deconfined quantum critical points in the long-range, anisotropic Heisenberg chain. With matrix product state simulations, we show that the model undergoes a continuous phase transition from a valence bond solid to an antiferromagnet. We extract the critical exponents of the transition and connect them to an effective field theory obtained from bosonization techniques. We show that beyond stabilizing the valence bond order, the long-range interactions are irrelevant and the transition is well described by a double frequency sine-Gordon model. We propose how to realize and probe deconfined quantum criticality in our model with trapped-ion quantum simulators.

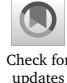

# 1 Introduction

The understanding of continuous phase transitions is largely based on Landau-Ginzburg theory [1]. The key concept is to characterize phases of matter by an order parameter that detects spontaneous symmetry breaking (SSB). Wilson has later introduced renormalization group ideas [2] to quantitatively account for the critical exponents which has led to the sophisticated Landau-Ginzburg-Wilson (LGW) paradigm.

In recent years, new classes of critical behavior that are not captured by this paradigm have been found. One example is the continuous transition between two SSB phases, that break different symmetries of the Hamiltonian, in two dimensional spin models [3,4] known as a Deconfined Quantum Critical Point (DQCP); for a recent review see [5]. Importantly, it has been established that such a transition is not captured within the LGW paradigm, as there is no a priori reason for why two continuous transitions of different order parameters should collapse at a single point. The DQCP can be rather understood by fractionalized constituents coupled to emergent gauge fields.

Recently, a transition that possesses many analogies with deconfined critical points has been proposed for a one-dimensional (1D) spin-1/2 model [6]. Contrarily to 2D where numerical evidence for a second order transition has been hard to obtain, the existence of powerful field theoretical and numerical tools in 1D has been used to gather strong evidence for a continuous transition [6–8]. Additionally, variants of the Lieb-Schultz-Mattis theorem [9] forbid the existence of a trivial phase in their model, thus rendering a conventional phase transition impossible. Following this work, DQCPs have been characterized in various other 1D models as well [10–14].

In this work, we study 1D DQCP in a long-range anistropic Heisenberg model with power-law decaying spin-spin interactions, that has been recently experimentally realized in a trapped ion quantum simulator [15]. We study the DQCP both analytically with field theoretic techniques and numerically with matrix product states. In our model the continuous transition between a valence bond solid (VBS) and an antiferromagnetic (AFM) Néel ordered phase is tuned by the power-law exponent of the long-range interactions and by the spin anisotropy. We employ bosonization techniques in (1+1)-dimension to show that the transition is described by an effective double frequency sine-Gordon field theory, which predicts an emergent U(1) symmetry at the critical point. Using tensor network simulations we extract the phase boundary and critical exponents of the transition. In accordance with the effective theory we find that the order parameters decay algebraically at the critical point with certain predicted relations between the critical exponents. Furthermore, the transition is characterized by a central charge $c = 1$. We propose experimental protocols for trapped ions to prepare the DQCP ground state and show how the emergent symmetry at the DQCP can be accessed through snapshots of the order parameters [13].

# 2 The model

In this work, we study a long-range, anisotropic Heisenberg chain described by the Hamiltonian

$$H_{\text{LR}}(\alpha, \Delta) = \sum_{j<i} \frac{J}{|i-j|^{\alpha}} \left[ S_i^x S_j^x + S_i^y S_j^y + (1+\Delta) S_i^z S_j^z \right], \tag{1}$$

where $\alpha$ is the power-law exponent of long-range interactions, $\Delta$ denotes the anisotropy in z-direction and $J > 0$ indicates the overall energy scale of the model. Throughout the following discussion we will fix $J = 1$ as unit of energy unless stated otherwise. Before we turn to a more detailed analysis of the phase diagram shown in Fig. 1 (a), let us discuss some general arguments for the limiting cases of $H_{\text{LR}}$.

We first consider the limiting cases of the long-range exponent $\alpha$. For $\alpha \to \infty$ the model reduces to the conventional XXZ chain, whose ground state for $\Delta = 0$ is a gapless Luttinger liquid and for any finite anisotropy $\Delta > 0$ is Néel ordered in z-direction [16]. For $\alpha \to 0$ all spins interact equally with each other $H_{\mathrm{LR}}(\alpha \to 0) = \frac{J}{2}\vec{S} \cdot \vec{S} + \Delta \mathcal{S}^z \cdot \mathcal{S}^z$, where $\vec{S} = \sum_i \vec{S}_i$ is a collective spin of extensive magnitude. At the isotropic point, $\Delta = 0$, the energy spectrum is determined using representation theory [17]. In the thermodynamic limit the ground states are given by arbitrary singlet representations with total spin zero. From $[\vec{S} \cdot \vec{S}, \Delta \mathcal{S}^z \cdot \mathcal{S}^z] = 0$ we find that isotropic and anisotropic contributions are diagonalized simultaneously. Given that $\mathcal{S}^z \cdot \mathcal{S}^z$ has eigenvalue 0 for spin-0 states, the ground state of the anisotropic limit also has singlet character, independent of $\Delta$.

When considering the isotropic limit $\Delta = 0$, a prior analysis [17] found that the model undergoes a quantum phase transition from a valence bond solid (VBS) for small $\alpha$, that is stabilized from the arbitrary singlet states, to a gapless Luttinger liquid (LL) at a critical value $\alpha \approx 1.66$. By contrast, in the regime of dominating $\Delta$, the antiferromagnetic coupling in z-direction favors an AFM Néel ordering of the spins independent of $\alpha$.

Due to the different character of the ground state for $\Delta = 0$ and $\Delta \to \infty$ a phase transition is expected at some $\Delta_c(\alpha)$ in this model. In the large $\alpha$ regime, the LL-AFM transition is in the Berezinskii-Kosterlitz-Thouless universality class and arises for infinitesimal $\Delta$ [16]. In this work, the focus is on the regime of small $\alpha \lesssim 1.66$, where the two limiting cases are both spontaneously symmetry broken (SSB) phases. We argue that in this regime the model features a DQCP describing a continuous transition between the two ordered phases; the VBS phase at small $\Delta$ and the AFM phase at large $\Delta$.

## 3 Phase diagram

We use infinite-system Density Matrix Renormalization Group (iDMRG) simulations [18–20] implemented within the TenPy library [21] and optimize over a Matrix Product State (MPS) in the thermodynamic limit to analyze the phase transition between the VBS and AFM phase. For our numerical study we approximate the powerlaw decaying interactions in Hamiltonian (1) by a sum of exponentials [22] which allows for an efficient Matrix Product Operator (MPO) representation. To obtain a good approximation of the interactions even for large distances between the spins we need to take sufficiently many exponentials into account, which limits the maximal accessible bond dimension to $\chi \lesssim 500$. At the largest bond dimensions we make use of a low-amplitude density mixer [23] to improve convergence. Moreover, we restrict our study to $\alpha \geq 1$ where the convergence of the DMRG algorithm is controlled.

To attain a first understanding of the transition, we fix $\alpha = 1.2$ and compute the dimerization and Néel order parameters

$$\Psi_{\mathrm{VBS}} = \frac{1}{N}\sum_n (-1)^n \left[\vec{S}_n \cdot \vec{S}_{n+1} - \vec{S}_{n+1} \cdot \vec{S}_{n+2}\right], \qquad M_{\mathrm{AFM}}^z = \frac{1}{N}\sum_n (-1)^n S_n^z, \qquad (2)$$

as a function of $\Delta$, see Fig. 1 (b). For large values of $\Delta$ we find a finite value of $M_{\mathrm{AFM}}^z$ and vanishing $\Psi_{\mathrm{VBS}}$ and vice versa for small $\Delta$. At $\Delta \approx 0.975$ both order parameters acquire a small jump, indicating a weak first order transition.

The (weakly) first order character of the transition is, however, a consequence of the finite bond dimension of the iMPS, which cannot resolve critical states with infinite correlation length. In our case, close to the critical point the approach is thus expected to encounter metastability due to the near degeneracy of low energy states with competing order. The jumps in the order parameters should therefore be regarded as finite residuals due to hysteresis close to the critical point, i.e., a bias of the optimized MPS towards the initial state used for

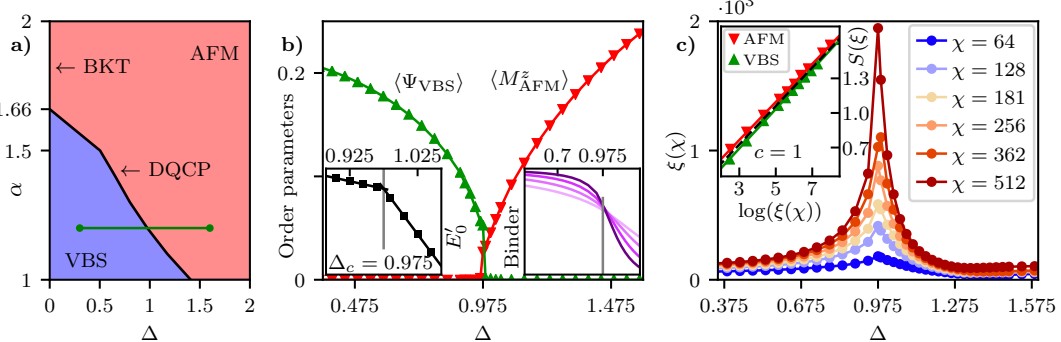

Figure 1: **Deconfined quantum criticality.** a) Phase diagram of the long-range, anisotropic Heisenberg chain. For $1.0 < \alpha \lesssim 1.66$ the model features a DQCP from a VBS to an AFM phase. b) Order parameters evaluated along the green line. c) Divergence of the correlation length $\xi$ with bond dimension $\chi$. Insets: b) Left: For an MPS with finite bond dimension, the transition is weakly first order leading to a kink in the MPS ground state energy from which we estimate the critical point. Right: The so-obtained critical value $\Delta_c \approx 0.975$ is consistent with a Binder cumulant analysis for $M_{\mathrm{AFM}}^z$. c) The critical entanglement entropy scales linearly with $\log \xi(\chi)$ yielding a central charge $c = 1$ consistent with field-theory predictions.

DMRG. While in [7] a sophisticated ramping protocol was necessary to control the hysteresis, we find that it is sufficient to choose different initial states for DMRG in our case; starting with the totally antiferromagnetic Néel state induces a finite residual of $M_{\mathrm{AFM}}^z$, using a singlet as initial state leads to a finite residual of $\Psi_{\mathrm{VBS}}$. In the following, we expand our protocol to address this problem and argue that the transition becomes continuous in the limit $\chi \to \infty$. In fact, we will see that the first order character is advantageous for determining the critical point $\Delta_c$ of the transition.

Due to the finite residuals, a good estimate for the critical point is hard to obtain by means of the order parameters. To improve precision, we use the weak first order character of the transition which leads to a (small) discontinuity in the energy at the critical point. A linear approximation of the energy on both sides of the critical point yields $\Delta_c^*(\alpha = 1.2, \chi = 724) \approx 0.975$, see left inset in Fig. 1 (b), in agreement with the behavior of the order parameters and a study of the Binder cumulant $\langle (M_{\mathrm{AFM}}^z)^4 \rangle / \langle (M_{\mathrm{AFM}}^z)^2 \rangle^2$ which is scale invariant at criticality [24] (Fig. 1 (b), right inset). The so obtained critical point $\Delta_c^*(\chi)$ varies with the bond dimension of the MPS. We emphasize, however, that for the largest bond dimensions used, the critical value is accurate up to $\delta \Delta_c \approx 5 \cdot 10^{-3}$.

With a good estimate for $\Delta_c$, a first indication of a continuous transition is found by evaluating the correlation length $\xi$ of the system, which is a property of the transfer matrix of the MPS [21] and diverges at a continuous phase transition. Such divergence cannot be captured directly due to the finite bond dimension of the MPS. Nonetheless, inside the gapped phases one rapidly approaches the true value of $\xi$ with increasing bond dimension resulting in a $\chi$-dependent cusp of increasing height at the transition as in Fig. 1 (c).

Further evidence for a continuous transition emerges from an analysis of the entanglement entropy at the critical point, which is simulated with both initial states, a Néel ordered as well as a singlet configuration. The scaling of the entropy with bond dimension follows $S(\chi) = \frac{c}{6} \log \xi(\chi)$ [25] with the central charge $c$, see inset in Fig. 1 (c). We find $c = 1$ for the central charge of the transition, which is consistent with the field theory presented in the next section, Sec. 4.

# 4 Effective field theory

From an analytical point of view the VBS-AFM transition can be understood by bosonizing the spin degrees of freedom, see e.g. [16]. The standard procedure consists of representing the spin operators by Jordan-Wigner (JW) fermions defined via

$$S_j^z = c_j^\dagger c_j - \frac{1}{2}, \qquad S_j^+ = (-1)^j c_j^\dagger \exp\left(i\pi \sum_{k<j} c_k^\dagger c_k\right), \tag{3}$$

where we employed a canonical transformation $\tilde{c}_j \mapsto (-1)^j c_j$ ensuring the correct definition of right- and left-moving fermions for our antiferromagnetic model later on. The JW fermions are then mapped to two bosonic fields $\phi(x)$, $\theta(x)$ through

$$\frac{c_j}{\sqrt{b}} \mapsto \psi(x) \equiv \psi_R(bj) + \psi_L(bj), \quad \text{with} \quad \psi_r(x) \equiv U_r \frac{e^{irk_F x}}{\sqrt{2\pi\gamma}} e^{-i[r\phi(x)-\theta(x)]}. \tag{4}$$

Here, $b$ is the lattice spacing and $\gamma$ a UV cutoff used to regularize the theory after taking the continuum limit. $\phi \in [0,\pi)$ and $\theta \in [0,2\pi)$ denote compact-valued bosonic fields following the algebraic structure encoded by $[\phi(x_1), \nabla\theta(x_2)] = i\pi\delta(x_2 - x_1)$. Moreover, $r = \pm 1$ for the right(left)-moving fermions $\psi_{R/L}$ that describe the low-energy excitations close to the two Fermi points $k = \pm k_F$. Lastly, the Klein factors $U_r$ account for creation and annihilation of right and left moving fermions and commute with the bosonic fields. Since they are irrelevant for the following discussion we drop them from here on. The non-interacting part of $H_{\text{LR}}$ is mapped to

$$H_0 = \sum_j (S_j^x S_{j+1}^x + S_j^y S_{j+1}^y) \mapsto -\frac{1}{2} \sum_j (c_j^\dagger c_{j+1} + \text{h.c.}), \tag{5}$$

which gives $k_F = \pi/(2b)$ at half filling, i.e. zero magnetization in z-direction.

We emphasize that obtaining an exact representation of Eq. (1) requires accounting for the Jordan-Wigner string contained in Eq. (3) in longer-ranged terms. Nonetheless, we will argue in the following that the essential physics is already captured by the short-range contributions to the Hamiltonian. For this we show that long-range terms represent irrelevant contributions in the Renormalization Group (RG) sense and that the presented results are independent of the exact cutoff chosen for the short range part as long as next-nearest neighbor interactions are taken into account.

**Short-range contribution** We first consider only nearest and next-nearest neighbor interactions. In this case, $H_{\text{LR}}$ reduces to the antiferromagnetic $J_1 - J_2$ XXZ chain, which has been studied in multiple works, see e.g. [10, 26]. In particular, we want to point to Ref. [10] investigating the VBS-AFM transition as an example of a DQCP. The Hamiltonian is first written in terms of JW fermions

$$H_{\text{XXZ}} = \sum_j \Bigg[ -\left[\frac{1}{2}(c_j^\dagger c_{j+1} + \text{h.c.}) - (1+\Delta)\left(\hat{n}_j - \frac{1}{2}\right)\left(\hat{n}_{j+1} - \frac{1}{2}\right)\right]$$

$$+ \frac{1}{2^\alpha}\left[\frac{1}{2}(c_j^\dagger(1 - 2\hat{n}_{j+1})c_{j+2} + \text{h.c.}) + (1+\Delta)\left(\hat{n}_j - \frac{1}{2}\right)\left(\hat{n}_{j+2} - \frac{1}{2}\right)\right]\Bigg]. \tag{6}$$

Taking into account only the lowest harmonics, the bosonized action reads

$$S_{\text{XXZ}}(\phi,\theta) = \int d\tau dx \left[\frac{i}{\pi}\partial_\tau\theta\nabla\phi + uK(\nabla\theta)^2 + \frac{u}{K}(\nabla\phi)^2 + \frac{\lambda}{(2\pi\gamma)^2}\cos(4\phi(x))\right], \tag{7}$$

where we introduce the Luttinger parameter K and coupling $\lambda = 2b^2\left[\frac{1}{2^\alpha}(3+\Delta)-(1+\Delta)\right]$. Additionally, the order parameters are represented by the expressions

$$
\begin{aligned}
M^z_{\text{AFM}} &\sim \frac{1}{\Omega}\int dx \cos(2\phi(x)), \\
\Psi_{\text{VBS}} &\sim \frac{1}{N}\sum_j (-1)^j \left[S^+_j S^-_{j+1} + \text{h.c.}\right] \sim \frac{1}{\Omega}\int dx \sin(2\phi(x)),
\end{aligned}
\tag{8}
$$

in terms of the bosonic fields. Note, that we project the VBS order parameter onto the (x,y)-plane, effectively reducing the SU(2) invariance to a U(1) phase choice. This is possible as the resulting operator behaves identically under the symmetry operations relevant to our model and thus serves as an alternative order parameter.

Having obtained the bosonized description for the model we start by relating the gapped phases to the strong coupling limits $\lambda \to \pm\infty$ of the action $S_{\text{XXZ}}$ in Eq. (7). For $\lambda \ll 0$, the field $\phi$ is pinned to $\phi = 0, \pi/2$ to minimize the action, thus leading to a finite expectation value of $M^z_{\text{AFM}}$. In contrast, $\lambda \gg 0$ restricts $\phi$ to take values of $\phi = \pi/4, 3\pi/4$ inducing a finite expectation value of $\Psi_{\text{VBS}}$. Under the premise that the cosine is relevant, the action $S_{\text{XXZ}}$ indeed describes a continuous VBS-AFM transition which happens when $\lambda$ changes sign. To determine the flow of the parameter $\lambda$ under a RG transformation, we evaluate the scaling dimension of the cosine in Eq. (7) for small $\lambda$. From the quadratic part in Eq. (7) one has $\dim[e^{im\phi}] = m^2 K/4$, $\dim[e^{in\theta}] = n^2/(4K)$ and thus $\dim[\cos(4\phi(x))] = 4K$. The resulting RG equation

$$
\frac{d\lambda}{dl} = (2 - \dim[\cos(4\phi)])\,\lambda,
\tag{9}
$$

hence flows to strong coupling whenever $2 - 4K > 0$. Hence, the cosine becomes relevant for $K < 1/2$. In our numerical results we indeed find $0.25 \lesssim K \lesssim 0.5$ at the dynamical critical point consistent with the cosine being relevant, see Sec. 5.

To see which terms from higher harmonics or additional interactions can appear in the short range part, let us first consider general restrictions imposed by symmetry of the Hamiltonian. Let us focus on the full Hamiltonian $H_{\text{LR}}$, which is invariant under translational symmetry $T_x$. Moreover, the full spin rotation symmetry $SU(2)$ is broken down to $U(1) \times \mathbb{Z}_2$, corresponding to rotations in the easy-plane and spin-flips along the easy-axis. For completeness let us mention that $H_{\text{LR}}$ is also invariant under time reversal symmetry $\mathcal{T}$ due to the absence of interactions containing an odd number of spin operators even though the symmetry does not play an important role in our considerations. The symmetry operators transform the spin operators as

$$
\begin{aligned}
T_x &: \vec{S}_j \to \vec{S}_{j+1}, \\
U(1) \equiv \prod_j e^{-i\alpha S^z_j} &: S^{x,y}_j \to \cos(\alpha)S^{x,y}_j \mp i\sin(\alpha)S^{y,x}_j, \quad S^z_j \to S^z_j, \\
\mathbb{Z}_2 \equiv \prod_j (2S^x_j) &: S^x_j \to S^x_j, \quad S^{y,z}_j \to -S^{y,z}_j.
\end{aligned}
\tag{10}
$$

After finding an approximation for the spin operators using Eq. (3) and Eq. (4), see also [16]

$$
\begin{aligned}
\frac{S^z_j}{b} &\mapsto S^z(x) = -\frac{1}{\pi}\nabla\phi(x) + \frac{(-1)^x}{\pi\gamma}\cos(2\phi(x)), \\
\frac{S^+_j}{\sqrt{b}} &\mapsto S^+(x) = \frac{e^{-i\theta(x)}}{\sqrt{2\pi\gamma}}[(-1)^x + \cos(2\phi(x))],
\end{aligned}
\tag{11}
$$

it becomes apparent that the symmetries act on the bosonic fields as

$$T_x \;:\; \phi \to \phi + \frac{\pi}{2}, \qquad \theta \to \theta\,,$$

$$U(1) \equiv \prod_j e^{-i\alpha S_j^z} \;:\; \phi \to \phi\,, \qquad \theta \to \theta + \alpha\,,$$

$$\mathbb{Z}_2 \equiv \prod_j (2S_j^x) \;:\; \phi \to -\phi + \frac{\pi}{2}, \qquad \theta \to -\theta\,. \tag{12}$$

Respecting the $U(1)$ symmetry allows only for terms involving differences of the field $\theta$ such as $\cos(\theta(y)-\theta(x))$. However, for small $|y-x|$ as in the case of short-range interactions such a term can be expanded up to first order, yielding

$$\cos(\theta(y) - \theta(x)) \approx \cos(\nabla\theta(x)|y - x|) \approx 1 + \mathcal{O}((\nabla\theta)^2)\,, \tag{13}$$

which does not add an additional $\theta$-dependence. Due to $\mathbb{Z}_2$ and translational symmetry, only higher cosine terms $\cos(4n\phi)$ can appear in the short-range part. Such a term scales as $\dim[\cos(4n\phi)] = 4n^2 K$ and is thus relevant for $K < 1/(2n^2)$. In particular, $n = 2$ being the next higher possible contribution implies that the relevant action remains unaltered as long as $K > 1/8$, and numerically we find $K$ to be always in this regime.

**Long-range contribution**   To capture the long-range interactions one uses the approximations for the spin operators in Eq. (11), which are inserted into the Hamiltonian in Eq. (1) to obtain the long range part of the action. First, we consider only non-oscillating terms, i.e. drop all contributions $\sim (-1)^x$ corresponding to higher harmonics. Doing so, easy-axis interactions along the z-direction contained in Eq. (1) only cause one additional term

$$S^z(x)S^z(y) \sim (1 + \Delta) \int dx\, dy\, \frac{1}{|x - y|^\alpha} \nabla\phi(x)\nabla\phi(y)\,, \tag{14}$$

which in Fourier space contributes as $\sim \int dq\, |q|^{\alpha+1}\phi(q)^2$. If we restrict ourselves to $\alpha > 1.0$, this term will be irrelevant with respect to the quadratic part of Eq. (7) that in Fourier space reads $\sim \int dq\, q^2\phi(q)^2$. Additionally, easy-plane contributions result in a term of the form

$$S_{\mathrm{LR}} \equiv \frac{1}{2}(S^+(x)S^-(y) + S^-(x)S^+(y))$$

$$\sim -\int dx\, dy\, \frac{1}{|x - y|^\alpha} \frac{b}{\pi\gamma}\Big[\cos\big(\theta(x) - \theta(y)\big)\cos\big(2\phi(x)\big)\cos\big(2\phi(y)\big)\Big]\,, \tag{15}$$

coupling the fields $\theta$ and $\phi$. This term scales as $\dim[S_{\mathrm{LR}}] = \alpha - 1 + 1/(2K) + 2K$ and will thus flow to strong coupling only if $K$ satisfies $\alpha < 3 - 2K - 1/(2K)$. Taking the Luttinger parameter $K$ as obtained from our numerical calculations below, we find again that throughout the considered parameter regime this term is irrelevant. We thus find that all long-range contributions are irrelevant in the RG sense. By contrast, it has been found that long-range ferromagnetic interactions drive the system toward a first order transition [14].

Interestingly, considering also the oscillating contributions arising from inserting Eq. (11) into Eq. (1) we find contributions surpressing certain types of order. The term

$$S_{\mathrm{XY}} \sim \int dx\, dy\, \frac{(-1)^{|x-y|}}{|x - y|^\alpha} \frac{b}{\pi\gamma} \cos(\theta(x) - \theta(y))\,, \tag{16}$$

arises from the xy-interaction in the easy plane and can induce continuous symmetry breaking of the U(1) symmetry for ferromagnetic couplings [27], where the oscillating prefactor

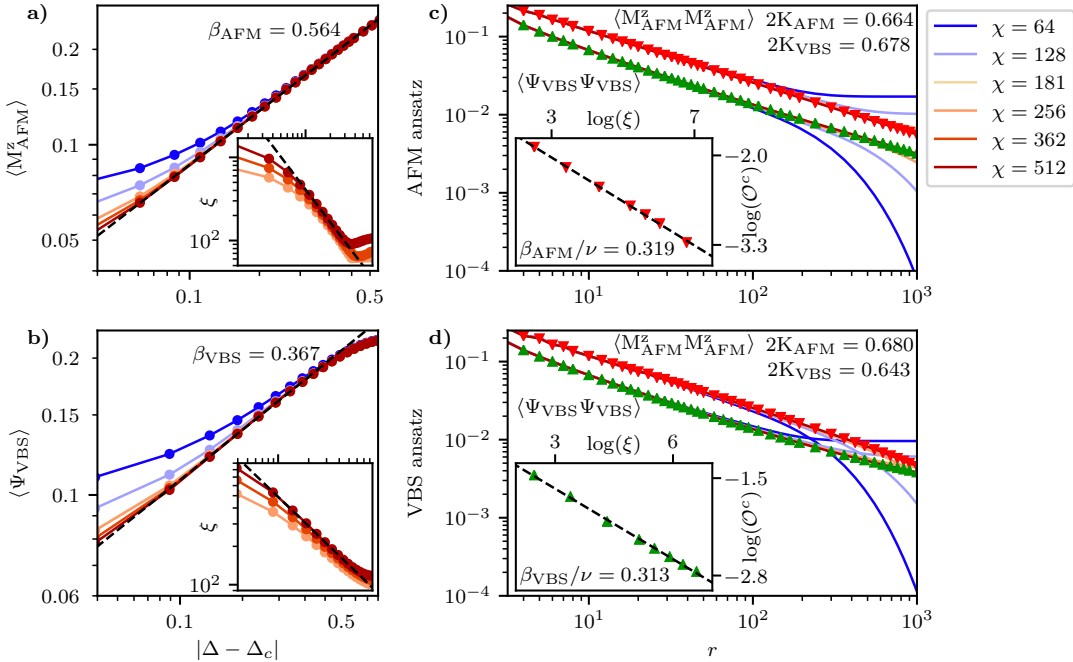

Figure 2: **Critical exponents of the AFM-VBS transition.** a)-b) The order parameters obey a power-law scaling in the vicinity of the critical point from which we extract the exponents $\beta_{\text{AFM}}$ and $\beta_{\text{VBS}}$. Insets: Scaling of the correlation length. As a consequence of the emergent enhanced symmetry group of the DQCP the critical exponents characterizing the algebraic decay of order parameter correlations are expected to coincide on both sides of the transition. Numerically this can be confirmed using two different initial configurations for the ground state search, c) an initial Néel as well as d) a singlet configuration. As a result of finite MPS bond dimensions accessible, correlations at large distances $r$ cross over from an algebraic to an exponential decay or to constant values given by the square of the order parameter residual. Insets: The order parameter residual at the critical point allows us to obtain numerical estimates for $\beta/\nu$, see main text for details. Results are plotted for $\alpha = 1.2$.

vanishes. Contrarily, with antiferromagnetic couplings, this term is always surpressed by the oscillating prefactor compared to the $\cos(4\phi(x))$ term from Eq. (7), thus leading to a dimerized phase instead of a continuously symmetry broken phase with finite expectation value of $S^x + S^y$ in the x-y plane.

Additionally, the term

$$S_{\text{ZZ}} \sim (1+\Delta) \int dx\, dy \, \frac{(-1)^{|x-y|}}{|x-y|^\alpha} \frac{b^2}{\pi^2\gamma^2} \cos(2\phi(x)) \cos(2\phi(y)), \tag{17}$$

arising from long range z-coupling along the easy-axis term prevents an $|\uparrow\uparrow\downarrow\downarrow\rangle$ (up-up-down-down) ordered phase which can occur in the short range $J_1 - J_2$ model when $J_2/J_1 > 1/2$, e.g. [10, 13]. This last condition follows from the simple argument that the energy $E_{\uparrow\uparrow\downarrow\downarrow}$ of the perfect $|\uparrow\uparrow\downarrow\downarrow\rangle$ state can become lower than that of the Néel state $E_{\text{N}}$. The same argument applied to the long range case gives $E_{\text{N}} < E_{\uparrow\uparrow\downarrow\downarrow}$ independent of $\alpha$, which is encoded in the $S_{ZZ}$ term. In terms of the short range action, this further shows that the next higher symmetry allowed $\cos(8\phi(x))$ term which accounts for the four-fold degeneracy of the $|\uparrow\uparrow\downarrow\downarrow\rangle$ state never becomes relevant.

| $\alpha$ | $\Delta_c$ | AFM | | VBS | |
|---|---|---|---|---|---|
| | | $\beta/\nu$ | $K$ | $\beta/\nu$ | $K$ |
| 1.5 | 0.503 | 0.401 | 0.401 | 0.385 | 0.394 |
| 1.4 | 0.655 | 0.358 | 0.369 | 0.375 | 0.385 |
| 1.3 | 0.801 | 0.351 | 0.350 | 0.34 | 0.354 |
| 1.2 | 0.975 | 0.319 | 0.332 | 0.313 | 0.321 |
| 1.1 | 1.175 | 0.29 | 0.303 | 0.283 | 0.309 |
| 1.0 | 1.409 | 0.251 | 0.278 | 0.259 | 0.299 |

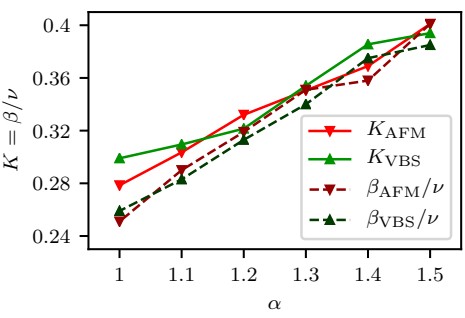

Figure 3: **Scaling of critical exponents.** Numerical values obtained for the critical exponents $\beta/\nu$ and $K$ in both phases. General scaling arguments predict $\beta/\nu = K$ as graphically shown on the right. Indeed the four lines aggree with good accuracy and we find $1/8 < K < 1/2$.

## 5 Critical exponents

Using the estimates for the critical points from Sec. 2 and the field theory from Sec. 4, allows us to determine the critical exponents for the transition. In particular, we are interested in the correlation length exponent $\nu$ as well as the order parameter exponents $\beta_{\text{AFM}}$ and $\beta_{\text{VBS}}$, characterizing the powerlaws

$$\xi(|\Delta - \Delta_c|) \sim |\Delta - \Delta_c|^{-\nu},$$
$$M_{\text{AFM}}^z(|\Delta - \Delta_c|) \sim |\Delta - \Delta_c|^{\beta_{\text{AFM}}},$$
$$\Psi_{\text{VBS}}(|\Delta - \Delta_c|) \sim |\Delta - \Delta_c|^{\beta_{\text{VBS}}}, \tag{18}$$

valid in the vicinity of the transition. The numerical results shown in Fig. 2 (a) and (b) indeed show the expected powerlaw behavior for the order parameters as function of $|\Delta - \Delta_c|$. Performing the same analysis for the correlation length $\xi(|\Delta - \Delta_c|)$, however, turns out to be more challenging in this model for the following reasons. On the one hand, due to finite values of the bond dimension $\chi$, the correlation length saturates close to the critical point, approaching a maximal value $\xi(\Delta \to \Delta_c) \to \xi(\chi)$ for each $\chi$. On the other hand, even in ordered phases convergence in the correlation length is hard to achieve due to long-range interactions. Consequently, we are left with only a narrow parameter window where the power law can be resolved, see insets of Fig. 2 (a) and (b). This causes the convergence of the correlation length to be rather slow and the numerical values of the critical exponent $\nu(\chi)$ to shift with bond dimension. Because of these numerical challenges, we do not use the correlation length exponents in our further analysis. We emphasize, however, that the order parameters converge reasonably with increasing bond dimension, see Fig. 2.

Field theory, moreover, predicts the order parameter correlations

$$C_{\text{AFM}}(r) = \langle M_{\text{AFM}}^z(x) M_{\text{AFM}}^z(x + r) \rangle, \qquad C_{\text{VBS}}(r) = \langle \Psi_{\text{VBS}}(x) \Psi_{\text{VBS}}(x + r) \rangle, \tag{19}$$

to decay algebraically at the critical point. In particular, the critical exponents associated to both correlations functions have to agree as they are functions of the same field $\phi$, i.e. $C_{\text{AFM}}(r) \sim C_{\text{VBS}}(r) \sim r^{-2K}$, implicitly also enforcing $\beta_{\text{AFM}} = \beta_{\text{VBS}}$. The above relation is obtained by using Eq. (8) and considering the respective correlations of the field $\phi$, see e.g. [16]. In Fig. 2 (c) and (d) we show numerical results for the order parameter correlations at the critical point. At large distances correlations of the order parameter either approach a constant value given by the square of the residual or decay exponentially, which is a result of

the finite correlation length. With increasing bond dimension we thus recover the expected algebraic decay characterized by the same critical exponents for both phases.

A more sophisticated scaling analysis, moreover, enables us to relate the critical exponents for correlation length and order parameter. Denoting the order parameter of interest as $M$ and the distance to the critical point as $|\Delta - \Delta_c| \equiv \delta$ the scaling law takes the form $M(\delta) \equiv \int \mathrm{d}x\, m(x;\delta) \sim \delta^\beta$ implying that the order parameter density $m(x;\delta)$ has scaling dimension $\beta/\nu + 1$. Here we use that $\delta$ has scaling dimension $1/\nu$ in proximity of the critical point enforced by the power law $\xi \sim \delta^{-\nu}$. When considering the algebraic correlations $C(r) \equiv \langle M(x - r/2)M(x + r/2)\rangle \sim r^{-2K}$ at the critical point, the transformation $C(r) \to C(r/\lambda) \sim \lambda^p C(r) \sim \lambda^{2\beta/\nu}C(r)$ implies $K = \beta/\nu$. Here, the last relation follows from dimensional analysis of the correlator.

In order to verify this relation numerically, we compute $\beta/\nu$ from finite-entanglement scaling of the order parameter residuals [7]. The general idea is to use the analogy to finite-size scaling in statistical mechanics: In a system with length $L$ an order parameter $M$ generally decays to zero as $L^{-\beta/\nu}$ at the critical point. If we now employ that the cutoff length in our system is given by the correlation length, the discontinuity $\mathcal{O}^c$ of the order parameters at the critical point is expected to scale as

$$\mathcal{O}^c_{\mathrm{AFM/VBS}} \sim \xi(\chi)^{-\beta/\nu}. \tag{20}$$

This scaling is indeed observed numerically and shown in the insets of Fig. 2 (c) and (d).

Numerical values of $\beta/\nu$ as well as $K$ obtained from AFM and VBS fixed-point initial states are summarized in Fig. 3. Reasonable agreement is obtained for the predicted relations. Small deviations are found in the vicinity of $\alpha = 1.0$, which we attribute to the fact that numerical convergence is more difficult to achieve in this regime.

## 6 Experimental prospects

Our model from Eq. (1) can be realized in trapped-ion quantum simulators using Floquet engineering [15, 28]. Collective vibrations of an ion crystal induced by off-resonant laser coupling mediate power-law decaying Ising interactions $H_{xx} = \sum_{i<j} J/|i-j|^\alpha S_i^x S_j^x$ with tunable exponent $\alpha$ [29]. By periodically applying global $\pi/2$-pulses around the z, x and y axes, effective dynamics with $H_{xx}$, $H_{yy}$ and $H_{zz}$ is stroboscopically realized. The respective interaction strengths are determined by the time period $\tau_a$ between the pulses with $a \in \{x, y, z\}$. By choosing $\tau_x = \tau_y$ and tuning $\tau_z \geq \tau_x$, the Hamiltonian Eq. (1) with tunable anisotropy $\Delta$ is realized using the same protocol. Experimentally, more refined Floquet protocols can improve Trotter errors and stability [15].

In order to experimentally study the ordered phases and the DQCP, the ground state has to be prepared adiabatically. The core procedure involves an extended Hilbert space of three states labelled by $\{g, \uparrow, \downarrow\}$ and has been introduced in a related context in Ref. [13]. First, the system is initialized in an auxiliary state $|\mathrm{GS}\rangle = \bigotimes_i |g\rangle_i$ of the ions. Then, the population of these auxiliary states is transferred to the physical spin states $|\uparrow\rangle, |\downarrow\rangle$. This can be achieved by adiabatically tuning the parameter $s \in [0, 1]$ in the Hamiltonian $H(s) = H_{\mathrm{LR}} + H_l(s)$, where $H_l(s) = \sum_i \Omega_{\mathrm{L}}(s)\big(|\sigma\rangle_i \langle g|_i + \mathrm{h.c.}\big) + \Delta_{\mathrm{L}}(s)\sum_i |g\rangle_i \langle g|_i$ couples the physical $|\sigma\rangle_i$ and the auxiliary states $|g\rangle$. A sufficiently slow ramp of the detunings $\Delta_{\mathrm{L}}(s)$ from a large negative to a large positive value while simultaneously turning on and off $\Omega_{\mathrm{L}}(s)$ transfers the population from $|g\rangle$ to the spin states $|\sigma\rangle$. In order to target the sector with zero total magnetization of $H_{\mathrm{LR}}$, required to characterize the DQCP, the adiabatic passage has to be performed by coupling alternate spin states $\sigma_i$ on even and odd lattice sites. This can be achieved by local addressing with different laser frequencies on even (odd) sites, respectively. The protocol is optimal when the time dependent parameters $\Omega_{\mathrm{L}}(s)$ and $\Delta_{\mathrm{L}}(s)$ are chosen such that the energy gap is minimal

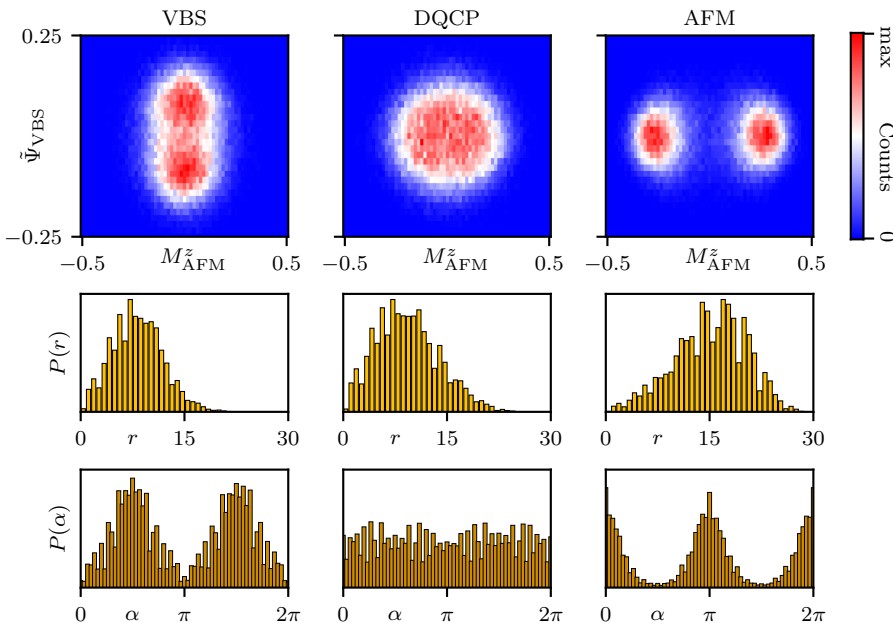

Figure 4: **Snapshots of the order parameters.** Joint probability distribution of the order parameters $\tilde{\Psi}_{\mathrm{VBS}}$ and $M^z_{\mathrm{AFM}}$. Top Row: Expectations values for both order parameters measured with respect to the ground state within the VBS (first column), the AFM (third column), as well as at the critical point (second column). We find a discrete peak structure of the distribution inside the ordered phases (AFM, VBS) reflecting the symmetry broken ground states. By contrast, at the critical point the obtained distribution is rotationally invariant indicating the emergent U(1) symmetry of the DQCP. The symmetry broken nature as well as the emergent symmetry at criticality are also reflected in the radial (middle row) and angular (bottom row) profile of the measurement outcomes.

at $H(s = 1)$. The gap is finite in the symmetry broken phases and closes algebraically with system size at the DQCP and determines the time scale of adiabatic preparation.

After preparing the state, the order parameters of the distinct phases need to be measured. A simultaneous measurements of the non-commuting AFM and VBS order parameters is in general not possible. Exploiting the symmetry of the problem we can, however, alternatively use the operator $\tilde{\Psi}_{\mathrm{VBS}} = 1/N \sum_i (\sigma^z_i \sigma^z_{i+1} - \sigma^z_{i+1} \sigma^z_{i+2})$ to characterize the VBS state by projecting the singlet operator onto one spin direction [13]. From that the full counting statistics of the joint distribution function is accessible. As an example, Fig. 4 shows the joint probability distribution of the order parameters for a unit cell of $N = 128$ sites taken from the iMPS ground state [30]. We fix $\alpha = 1.2$ and choose values of $\Delta = 0.4, 0.975, 1.55$ inside the VBS phase, at the critical point and inside the AFM phase and perform $3 \cdot 10^4$ measurements for each parameter.

Inside the ordered phases two peaks corresponding to the symmetry broken ground states indeed emerge as expected. We emphasize that although being less pronounced, breaking of rotational symmetry of the order parameter distribution is also observed closer to the DQCP. The rotational invariance of the joint distribution at the critical point is a clear indicator of an emergent U(1) symmetry akin to a deconfined quantum phase transition. This has to be contrasted with a conventional coexistence phase present at first order transitions, which would be characterized by a distribution containing four distinct peaks invariant under the discrete symmetry $\mathbb{Z}_2 \times \mathbb{Z}_2$. Moreover, we point out that the results at the DQCP in the second column of Fig. 4 are independent of how we initialize the DMRG ground state search.

# 7 Conclusions and outlook

We have investigated the zero temperature phase diagram of a long-range anisotropic Heisenberg model with power law decaying interactions. For sufficiently small power-law exponents $\alpha < 1.66$ we find a deconfined quantum critical point (DQCP) between a dimerized valence bond solid (VBS) and an antiferromagnetic Néel phase (AFM) which is forbidden in standard Landau-Ginzburg theory.

By employing bosonization techniques we show that the transition is described by an effective sine-Gordon theory with double frequency which arises from the nearest and next-nearest neighbor interactions. This effective theory is a consequence of the symmetries of our model and generally holds as long as the symmetry group remains unaltered. We furthermore argue that in the considered regime, the longer-ranged interactions, beyond next-nearest neighbors, become irrelevant and thus do not contribute to the effective theory.

Furthermore, density matrix renormalization group (DMRG) simulations are employed to analyze the transition numerically. The phase boundary hosting the DQCP is determined by analyzing the weak first order crossing in the energy, which arises from the finite matrix product state bond dimension, and a study of the Binder cumulant. In accordance with the effective theory a central charge $c = 1$ is determined through finite entanglement scaling and the critical exponents of the transition are determined. As predicted by the effective theory, both order parameters decay algebraically at the critical point with matching exponent related to the Luttinger parameter of the theory. We moreover show, how the DQCP can be characterized experimentally using trapped ions.

For future work, it will be interesting to study the finite temperature phases of this model. Even though the system is one dimensional, long-range interactions can stabilize ordered phases at finite temperatures [31,32]. Understanding and characterizing the fate of the DQCP in this regime could be an interesting future direction. Moreover, exploring the dynamics of excitations [15] could provide additional insights into the structure of the DQPT.

# Acknowledgments

We thank Ruben Verresen for stimulating discussions.

**Funding information** We acknowledge support from the Deutsche Forschungsgemeinschaft (DFG, German Research Foundation) under Germany's Excellence Strategy–EXC–2111– 390814868, TRR 360 - 492547816 and DFG grants No. KN1254/1-2, KN1254/2-1, the European Research Council (ERC) under the European Union's Horizon 2020 research and innovation programme (grant agreement No. 851161), as well as the Munich Quantum Valley, which is supported by the Bavarian state government with funds from the Hightech Agenda Bayern Plus.

**Data and code availability** Numerical data and simulation codes are available on Zenodo upon reasonable request [33].

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
