# Peer review of "Deconfined Quantum Criticality in the long-range, anisotropic Heisenberg Chain"

_SciPost Physics Core, doi:SciPost Phys. Core 7, 008 (2024)_

## Round 1 · Referee Report · Anonymous · 2023-12-19

Strengths

1. Clear presentation of the results.
2. Motivation comes from recent experimental realization of spin chain with long-range interactions.
3. Numerical results complemented and supported by field theoretical techniques.
4. Agreement between critical exponents for different order parameters provides compelling evidence for a 1D deconfined quantum critical point.

Weaknesses

1. The results do not offer any fundamentally new insight because the same transition has been studied in very closely related models, both by bosonization and by DMRG.
2. The section about experimental prospects essentially repeats the proposal and analysis from Ref. [12].

Report

Anisotropic spin chains with frustrated interactions beyond nearest neighbors have been studied in recent years in the context of 1D analogs of deconfined quantum criticality; see e.g. Refs. [6] and [10]. In this context, the present manuscript investigates the continuous transition from antiferromagnetic (AFM) to valence bond solid (VBS) order in a spin chain model with power-law-decaying interactions that can be realized experimentally (Ref. [13]). The authors combine numerical (iDMRG) methods with a standard bosonization analysis to show that the transition is described by an effective field theory with central charge c=1 (a Luttinger liquid). They also extract the Luttinger parameter from its relation to critical exponents. Within the limitations associated with the finite bond dimension used in the iDMRG simulations, the results are consistent with the field theory predictions.

The sine-Gordon theory for the transition in Eq. (7) has been extensively discussed in the literature; see e.g. Refs. [10] and [12]. In fact, if the long-range interactions are truncated at next-nearest neighbors, the Hamiltonian in Eq. (1) reduces to the one studied by Mudry et al. in Ref. [10], where both bosonization and DMRG methods were used to characterize the transition. In section 4 the authors argue that the long-range contributions do not modify the essential physics as long as the exponent \alpha is large enough. Indeed, the AFM-VBS transition was also studied using very similar methods in a model with long-range interactions in Ref. [12]. In contrast with Ref. [12], where the exponent was fixed to correspond to dipolar and Van der Waals interactions, in this manuscript the authors vary the exponent between \alpha=1 and \alpha=2, but the results are qualitatively the same. The discussion in section 6 about detecting the emergent U(1) symmetry at the critical point using snapshots of the order parameters is also very similar to what was done in Ref. [12].

In my opinion, this manuscript does not present groundbreaking discoveries or breakthroughs in comparison with previous results in the literature. Therefore, it does not meet the expectations listed in the acceptance criteria of SciPost Physics, but it could be published in SciPost Physics Core.

Requested changes

1. When discussing the limits of the model in Sec. 2, I suggest citing papers that considered the isotropic limit before Ref. [15], for instance Parreira et al., J. Phys. A 30, 1095 (1997); Laflorencie et al., J. Stat. Mech. P12001 (2005).
2. In the paragraph above Eq. (6), I believe the authors meant to say that they considered nearest and next-nearest neighbor interactions (not only next-nearest).
3. Please define K as the Luttinger parameter right below Eq. (7), where it first appears.
4. Please write the value of \alpha in the caption of Fig. 2.

---

## Round 1 · Referee Report · Anonymous · 2023-12-20

Strengths

Scaling analysis is carefully done to show convincing numerical evidence for deconfined quantum criticality in the long-range, anisotropic Heisenberg chain.

Report

This paper discusses deconfined quantum criticality between the valence bond solid (VBS) ground state and the antiferromagnetic (AFM) ground state in the long-range anisotropic Heisenberg chain, using matrix product state simulations and bosonization methods.
The phase diagram is obtained from the numerical simulations in the two-parameter space of the exponent $\alpha$ of the long-range interaction ($\propto 1/r^\alpha$) and the exchange anisotropy $\Delta$, and the phase boundary between the VBS and AFM phases is determined for $\alpha>1$. Through a careful scaling analysis with varying bond dimensions $\chi$, the phase transition is shown to be continuous, with the central charge $c=1$ obtained from the scaling of entanglement entropy.
The low-energy effective theory for the phase transition is derived using bosonization, reproducing the previous theory in Refs. [10,24]. An important new result in this derivation is that the long-range part of the exchange interactions (beyond the next-nearest-neighbor interactions discussed in [10,24]) is irrelevant in the renormalization-group sense.
The critical exponents $\beta_{\rm VBS}$ and $\beta_{\rm AFM}$ for the VBS and AFM order parameters and the correlation length exponent $\nu$ are also estimated from the numerical simulations. While it turns out to be difficult to determine the individual scaling exponents with good accuracy, the scaling analysis of the residual order parameters at critical points due to finite bond dimensions allows the authors to successfully obtain a good agreement between the critical exponents ratio $\beta/\nu$ and the Luttinger parameter $K$ in support of the effective theory.
Finally, a proposal is made for realizing and observing, with trapped-ion quantum simulators, the deconfined quantum criticality in the theoretical model studied in this paper.
The results of this paper summarized above are reasonable and valid, and the proposal might stimulate experimental studies. The manuscript is well written and easy to read. I can therefore recommend this paper for publication in SciPost Physics.

---

## Round 2 · Author Response

We thank the referees for reviewing our work. Following the suggestion of Referee 1 we transfered our work to SciPost Physics Core. The requested changes are addressed in the revised version of the manuscript and our replies to the referees.

---

## Round 2 · List of Changes

- Clarified that we consider *nearest* and next-nearest neighbor interactions above Eq. (6).
- Defined K as Luttinger parameter immideately after Eq. (7).
- Added value of $\alpha$ in the caption of Fig. 2.

---

## Editorial Decision

published